# Synthesis and Sorption Properties towards Sr-90 of Composite Sorbents Based on Magnetite and Hematite

**DOI:** 10.3390/ma13051189

**Published:** 2020-03-06

**Authors:** Andrei Egorin, Eduard Tokar, Anastasia Kalashnikova, Tatiana Sokolnitskaya, Ivan Tkachenko, Anna Matskevich, Evgeny Filatov, Larisa Zemskova

**Affiliations:** Institute of Chemistry, Far Eastern Branch, Russian Academy of Sciences, prospect 100-letiya Vladivostoka, 159, 690022 Vladivostok, Russia; d.edd@mail.ru (E.T.); Kalashnikova1997n@yandex.ru (A.K.); sokolnitskaya@ich.dvo.ru (T.S.); tkachenko@ich.dvo.ru (I.T.); mysmatskevich@mail.ru (A.M.); Evgeni_Filatov735@mail.ru (E.F.); zemskova@ich.dvo.ru (L.Z.)

**Keywords:** composite materials, polymers, iron oxides, strontium, adsorption, ion-exchange

## Abstract

The article describes the synthesis of composite sorbents by immobilizing iron oxide in a polymer matrix with subsequent hydrothermal treatment at a temperature of 175 °C. The sorbents based on magnetite and hematite were synthesized, their magnetic properties and phase composition were evaluated, and the iron content was determined. Sorption characteristics of the composites towards microconcentrations of Sr-90 radionuclide in solutions with different mineralization and pH were investigated. It was shown that the sorbent based on magnetite was the most efficient. In alkaline media with pH above 11, the composite sorbent based on magnetite exhibited increased selectivity towards Sr-90 and proved to be suitable for application under dynamic sorption conditions with subsequent desorption of the radionuclide with a solution of HNO_3_.

## 1. Introduction

Currently, scientists working in the field of radiochemistry are facing a challenge of creating efficient materials for treatment of the clarified part of heterogeneous liquid radioactive waste (LRW) streams accumulated in the process of the nuclear weapon production in order to remove Sr-90. The chemical composition of the clarified part of heterogeneous LRW is mainly represented by Na^+^ ions (more than 1 mol/L) in the forms of nitrites and nitrates, as well as OH^−^ (more than 0.5 mol/L) [1,2,3].

Similar heterogeneous LRW with analogous chemical composition in the amount of 14.5 × 10^3^ m^3^ with the activity of 74 × 10^6^ Ci are stored at the territory of the Russian Federation, namely, at the Mayak Nuclear Plant [4,5]. The composition of these waste streams is mainly represented by Na^+^ ions (3.0–4.5 mol/L) [6]; their activity is mainly determined by the presence of Cs-137 and Sr-90 radionuclides.

Substances capable of extracting Sr-90 from highly mineralized alkaline solutions include iron oxides (hydroxides), as well as composite sorbents based on them. The advantages of these materials are low cost, ease of production, high efficiency, and non-toxicity.

Sorption of strontium on iron oxides proceeds due to formation of surface complexes described by the diffuse double layer model [7]. The mechanism of Sr sorption on magnetite was described in detail in [8]; the authors described the sorption by the reaction >FeOH + Sr^2+^ ↔ >FeOHSr^2+^, for which the logarithm of the equilibrium constant was equal to 2.7 ± 0.3. It was shown that the sorption of strontium by magnetite was strongly affected by carbonates, which was related to the formation of strong carbonate–strontium complexes. Karasyova et al. [9] investigated the effect of temperature on strontium sorption by hematite. According to the calculations, the sorption process, depending on pH and concentration, was followed by the formation of the surface complexes >FeOSr^+^ and >FeOSrOH. The formation of the indicated complexes occurred at a pH above 9; however, the increase in the temperature of the model solution led to the shift of the complex formation reaction to the weakly alkaline region.

To remove strontium, a method was suggested in which the formation of mixed iron oxides occurred (in situ) when Fe(II)/(III) iron salts were added to supernatants with co-deposition of Sr-90 radionuclide with iron hydroxide, as well as Pu, Np, Am, and U [1,3].

Over the last decade, composite sorption materials based on iron oxides were developed for the removal of strontium; for instance, composite sorbents based on chitosan [10,11], graphene oxide [12], silicate [13], aluminosilicate [14,15], alginate [16], triazine polymer [17], etc., were suggested. The promising sorbent was synthesized for removal of arsenic from liquid media. The material comprised particles of iron oxide homogeneous dispersed in the acrylamide-based cryogel [18]. The resulting material was characterized with high stability and high adsorption capacity.

Hansen et al. [19] suggested a sorbent comprising sand modified with iron oxide (III) for removal of Sr from LRW of the Hanford Tank Wastes. This sorbent enables one to treat about 50,000 bed volumes of LRW and comprises a rather advantageous material due to its low cost.

However, modification of the surface of organic and inorganic matrices with deposition of iron oxides on the surface could result in a loss of iron oxide due to mechanical impact, as well as peptization, which could lead to the decrease in the efficiency of a sorbent. To eliminate these disadvantages, iron oxide can be immobilized in the bulk of a polymer matrix permeable to Sr^2+^ ions, which allow production of qualitatively new sorbents. Here, a major part of sorption materials is represented by polymers filled with metal oxides. Introduction of metal oxide particles into a polymer matrix enables one to protect oxide particles from the tendency towards aggregation and to provide the working pressure in a column [20]. Some examples of such hybrid sorbents were arsenic sorbents fabricated on the basis of a strong-base anion exchanger [20,21,22] and magnetic composites produced by precipitation of mixed iron hydroxides in hypercrosslinked or mesoporous polystyrene-type matrices [23] or in a matrix of a natural biopolymer-chitosan [11,24].

From a technical point of view, the most suitable is the composite material presented in [25]. The sorbent comprising Fe_3_O_4_ nanoparticles immobilized by *Penicillium* sp. mycelium, can be used to remove Sr(II), Th(IV), and U(VI) from liquid media. The composite material is characterized with high stability and adsorption capacity and can be used in sorption–desorption cycles. Despite its advantages, the composites synthesis is quite difficult and includes *Penicillium* sp. mycelium growing during a long time. Besides, the authors did not provide the data about using this material under dynamic sorption conditions.

Universal strongly acidic gel type cation-exchangers of the KU 2-8 and Purolite C100E grades extensively applied for removal of various cations [26,27], can be used as matrices of composite sorbents. The polymer structure of cation-exchangers is composed of styrene–divinylbenzene containing sulfonic acid functional groups. The advantages of these ion exchangers as composite sorbents matrices are in the permeability, high chemical and mechanical stability, and spherical granular form that makes convenient manipulations with the fabricated materials.

The objective of the present work was to synthesize and study the sorption characteristics of sorbents based on magnetite and hematite immobilized in the bulk of an ion-exchange resin polymer KU 2-8 [28].

## 2. Materials and Methods

### 2.1. Materials

Iron(III) chloride (FeCl_3_ × 6H_2_O), iron(II) sulfate (FeSO_4_ × 7H_2_O), ammonium hydroxide (NH_4_OH), sodium hydroxide (NaOH), and sodium nitrate (NaNO_3_) of the chemically pure grade were purchased from JSC Nevareaktiv, Russia and used without any additional purification. The batch of Sr-90 radionuclides in 1 M HCl solution was purchased at the Leypunsky Institute of Physics and Power Engineering. The cation exchanger KU 2-8 was purchased from the TOKEM company, Russia [28]. Prior to the work start, the cation-exchanger was washed with 1 M HNO_3_ solution under dynamic conditions; thereafter, it was washed from acid residues with distilled water and stored in a flask with a ground stopper.

### 2.2. Synthesis of Magnetite

The synthesis of magnetite powder (Mag) was carried out as follows: 2 mL of H_2_SO_4_ solution of a concentration of 10 mol/L was added to 20 g of FeSO_4_ × 7H_2_O (0.072 mol) and dissolved in 100 mL of distilled H_2_O; then 100 mL of FeCl_3_ solution with a concentration of 0.72 mol/L (0.072 mol) was added to the resulting Fe(II) solution. A solution of NH_4_OH of a concentration of 10 mol/L was added to the Fe(II)/Fe(III) solution dropwise under continuous stirring until a sharp increase in the viscosity due to the formation of a dark green precipitate at pH of 10 ± 1. The resulting precipitate was separated from the solution by centrifugation for 10 min (3000 rpm), after which it was transferred to a Teflon autoclave glass and filled with 50 mL of NH_4_OH solution of a concentration of 0.1 mol/L. The hydrothermal treatment was carried out for 24 h at a temperature of 175 °C, after which the reactor was gradually cooled down to room temperature. The resulting black precipitate was washed with distilled water in a Buchner funnel and dried at 85 °C until constant weight.

### 2.3. Synthesis of Hematite

The synthesis of hematite powder (Hem) was carried out as follows: a solution of 10 mol/L NH_4_OH was added dropwise to 200 mL of 0.72 mol/L Fe(III) solution until a thick brown precipitate was formed (pH 9–11). The precipitate was washed with distilled water on a Buchner funnel from ammonia residues and then transferred to a Teflon autoclave glass, and 100 mL of a 0.1 mol/L HCl solution was added. The hydrothermal treatment was performed for 24 h at 175 °C, and the resulting product was separated from the solution by centrifugation at 4500 rpm, and, thereafter, dried at 85 °C until constant weight. The resulting precipitate had a red color and was easily peptized.

### 2.4. Synthesis of Magnetite Composite Sorbent Based on KU 2-8 and Magnetite (KU–Mag)

Fe(II)/Fe(III) solution was prepared as follows: 3 mL of H_2_SO_4_ solution of a concentration of 10 mol/L and a FeSO_4_ × 7H_2_O sample with a weight of 1.4 g (5 × 10^−3^ mol) were added to 25 mL of distilled H_2_O. After dissolution of FeSO_4_ × 7H_2_O, 25 mL of FeCl_3_ solution of a concentration of 0.2 mol/L (5 × 10^−3^ mol) was added. The swollen KU 2-8 resin of a volume of 50 mL was brought into contact with a solution of Fe(II)/Fe(III) for 12 h under continuous stirring on an orbital shaker. After a specified time, the resin was separated from the solution and washed with distilled water in a Buchner funnel.

To form the magnetite phase, 50 mL of distilled water was added to the resin in the form of Fe(II)/Fe(III), and a 10 mol/L NH_4_OH solution was added dropwise under continuous stirring. The addition of the NH_4_OH solution was stopped when the pH of the solution reached 10 ± 1, and this value was maintained for 30 min. Thereafter, the resin was washed from magnetite particles formed outside the resin bulk by means of decantation and transferred to the Teflon autoclave glass. A solution of NH_4_OH with pH 11–14 was added to the resin and autoclaved for 12 h at a temperature of 175 °C. The resulting resin had a black color.

### 2.5. Synthesis of Hematite Composite Sorbent Based on KU 2-8 and Hematite (KU–Hem)

Synthesis of sorbent was based on KU 2-8 and hematite (KU–Hem). The swollen KU 2-8 resin of a volume of 50 mL was brought into contact with a 0.2 mol/L FeCl_3_ solution of a volume of 50 mL for 12 h under continuous stirring on the orbital shaker. After a specified time, the resin was separated from the solution and washed with distilled water in the Buchner funnel.

After the KU 2-8 alkalizing by NH_4_OH solution of a concentration of 10 mol/L until pH 10, the resin was washed via decantation and placed in the autoclave Teflon glass. Then, 0.1 mol/L HCl solution was added to the resin and processed in the autoclave for 12 h at a temperature of 175 °C.

### 2.6. Characterization

The iron content in the composite sorbents was determined as follows. The composite was ground and filled with a concentrated HCl solution. The resulting pulp was heated up to 70 °C under continuous stirring for 24 h. Thereafter, the solution was separated from the pulp, which was additionally washed with distilled water. The resulting solutions were combined and analyzed on the iron content using the atomic absorption spectroscopy method. The quantity of iron washed away from composites was known, and the phase content Fe_3_O_4_ and Fe_2_O_3_ was calculated.

The size (*D*) of iron oxide particles in the original powder and in composites was determined from the XRD data using the Scherrer equation (Equation (1)):(1)$$D=\frac{0.94\lambda }{\beta_{1/2}COS\theta }$$
where *β_1/2_* is the line broadening in radians, *Θ* is the Bragg angle, and *λ* is the X-ray wavelength (CuK-α 1.5406 Å).

### 2.7. Procedure of Sorption Characteristics Study

We used model solutions, the composition of which are shown in Table 1. The model solution No. 4 simulated a clarified fraction of heterogeneous LRW.

Prior to the experiments, the model solutions were labeled with the radionuclide Sr-90 (1000 Bq/mL). The removal of strontium was performed under static conditions when the sorbent was continuously mixed with a model solution in a 10 mL polypropylene cylinder at a speed of 20–30 rpm in a vertical rotary shaker; the mixing lasted for 7 days. Three parallel samples were used for each experiment, including a test experiment without a sorbent. After the specified time, the model solution was separated from the sorbent by centrifugation at a speed of 4500 rpm for 15 min, and then the residual activity was determined.

The distribution coefficient for Sr was calculated according to Equation (2) as follows:(2)$$K_{d}=\;\frac{A_{0\;}-\;A_{eq}}{A_{eq}}\;\times \frac{V_{st}}{m}$$

The sorption of Sr-90 under static conditions (*S_st_*, %) was calculated using Equation (3) as follows:(3)$$Sorption=\left(1-\frac{A_{0}}{A_{eq}}\right)\times 100$$
where *A_0_* is the initial activity of the model solution (Bq/mL), *A_eq_* is the equilibrium activity of the liquid medium (Bq/mL), *V_st_* is the volume of the liquid medium during sorption under static conditions (mL), and *m* is the mass of the sorbent weight sample (g).

The efficiency of Sr-90 removal under dynamic conditions was determined using the model solution No. 2. For this purpose, 2 mL of the KU–Mag sample with a grain size of 0.5–1.0 mm was placed in a column of an internal diameter of 1 cm. The model solution labeled with Sr-90 radionuclide (1000 Bq/mL) was passed through the column at a rate of 10 mL/h. The filtrate was collected in fractions of 8–15 mL and the residual activity was determined. Desorption of Sr-90 was performed using a 1 M HNO_3_ solution, which was passed through the column with the sorbent at a rate of 2 mL/h; the eluate was collected in fractions of 2.5–5.0 mL, and then the desorption efficiency was calculated.

The decontamination factor of Sr-90 (*DF*) under dynamic conditions was calculated using Equation (4) as follows:(4)$$DF=\frac{A_{0}}{A_{f}}$$

The desorption of Sr-90 (*Des*, %) under dynamic conditions was calculated using Equation (5) as follows:(5)$$Desorption=\frac{\sum_{1}^{n}{\left(A_{des}\times V_{des}\right)}_{i}}{\left(A_{0}\times V_{S}\right)-\left(A_{SF}\times V_{S}\right)}$$
where *A_0_* is the initial activity of the model solution (Bq/mL), *A_f_* is the residual activity of the *i*-th fraction of the filtrate (Bq/mL), *A_des_* is the activity of the *i*-th fraction of the eluate (Bq/mL), *A_SF_* is total residual activity of the model solution after filtration (Bq/mL), *V_des_* is the volume of the *i*th fraction of the eluate (mL), *V_S_* is total volume of the model solution passed through the column at the stage of sorption (mL), *n* is total number of the eluate fractions, and *i* is the number of a fraction of the eluate.

### 2.8. Analysis

The activity of the model solutions towards Sr-90 was determined using a liquid scintillation alpha–beta radiometer of the spectrometric Tri-Carb 2910 TR device (Perkin Elmer, USA). The minimum detectable activity was ~1 Bq/L with natural background 2.1 decay per minute. The measurement error depending of extinction coefficient (blanking ratio) was 5%.

The X-ray diffraction analysis was performed using a D8 ADVANCE diffractometer. X-ray patterns of the samples were recorded in the range of angles 2θ of 3–85° with an increment of 0.02° at a count of 0.6 s. Identification of the phase composition was performed using the QualX software (version 2.24) and the Crystallography Open Database (2019-06-27). The results were processed using the SciDavis software (version 1.23). The confidence interval for experimental values with probability 0.95 (standard deviation × 1.96) was calculated and is given below:

(1) For a series of parallel experiments (sampling) we used one-sample t-test.

(2) For cases when use of sampling was impossible the confidence interval was calculated with Equation (6) as follows:(6)$$SD=\left(\frac{\sqrt{n}}{t}\times 1.96\right)+m_{e}$$
where *n* is the quantity of registered impulses, *t* is the measurement time (min), and *m_e_* is the instrument measurement error. 

Magnetization of the samples was measured using a MPMS XL-7 SQUID magnetometer (Quantum Design) in the field range ±10,000 Oe at a temperature of 300 K. A measurement step equaled to 100 Oe in the range from 0 to ±2000 Oe, and to 500 Oe in the range from ±2000 to ±10,000 Oe.

## 3. Results

Figure 1 shows a simplified diagram of the formation of the iron oxide phase in the ion-exchange resin matrix. When alkali was added to the ion-exchanger in the Fe form, iron hydroxides were formed directly in the bulk of the ionite grain, and the color of the resin changed to brown when the ion-exchanger was in the Fe(III) form, or black when it was in the Fe(III)/Fe (II) form. Note that during the synthesis of magnetite in air, not only the magnetite phase (Fe_3_O_4_) was formed, but also isomorphic mixtures of the magnetite/maghemite phases.

The iron content and, accordingly, the mass of the hematite and magnetite phases for the composite sorbents were determined (Table 2). The iron content in the KU–Mag sample was more than two times higher than that of the KU–Hem.

Figure 2 shows the X-ray patterns of the pure magnetite and hematite powders (Figure 2a), the composite sorbents, and the initial KU 2-8 (Figure 2b). The Mag and Hem samples represented magnetite and hematite powders, respectively. In the KU–Mag and KU–Hem samples, the phases of iron oxides became weakly distinguishable, which was related to, first, the superposition of the amorphous polymer halo (Figure 2b, curve 5) and, second, the fact that the iron oxide phase was formed in the polymer matrix, presumably, in the form of fine crystallites.

Table 3 shows the values of iron oxide particle sizes calculated from the X-ray patterns for the initial powders and composite sorbents. According to the obtained data, the immobilization of iron oxides in the polymer matrix led to the decrease in the average size of particles. 

Figure 3 shows the dependencies of the magnetization of the sorption materials on the magnitude of the magnetic field H obtained at room temperature. Except for the pure hematite powder, the sorbents can be attributed to the group of soft magnetic materials. At the transition from the pure powder to the composite, the saturation magnetization value reduced significantly, which was related to the decrease in the mass fraction of iron oxide. Despite this fact, the KU–Mag sorbents retained their ferromagnetic properties, and the saturation magnetization values made it easy to separate the sorbents from the solution by means of magnetic separation. Moreover, the magnetically soft characteristics prevented the material from adhering during mixing. It is worth mentioning that in the case when iron oxide particles in the polymer are spaced apart and do not interact with each other, they must exhibit superparamagnetic properties, which must be experimentally manifested in the absence of saturation on a hysteresis loop. However, according to the magnetization curves shown in Figure 3c, the iron oxide particles were not completely isolated from each other and formed agglomerates.

Figure 4 shows diagrams of the dependence of K_d_ of Sr-90 on the content of Na^+^ ions in the model solutions with varied pH values. Sorption-selective characteristics of the initial powders and composite sorbents in a neutral solution of NaNO_3_ (pH 6.5–7.2) in the range of concentrations from 10^−3^ to 1.0 mol/L are shown in Figure 4a. The evaluation results of the efficiency of extraction of Sr-90 radionuclide from the alkaline solutions of NaNO_3_ + NaOH (pH 13) depending on the sodium concentration in the range 0.1–5.0 mol/L are shown in Figure 4b.

Since the sorption of Sr-90 in/on iron oxides is pH-dependent, the efficiency of radionuclide removal from the solutions with varied pH values was estimated (Figure 5). Figure 5a shows the dependence of the sorption-selective characteristics of the iron oxide powders and composite sorbents on the pH of the solution containing sodium (NaNO_3_ + NaOH) of a concentration of 0.1 mol/L. The efficiency of removing Sr-90 from the solutions with the sodium concentration of 0.5 mol/L depending on the pH is shown in Figure 5b.

In the case of use of the composite sorbents, the contribution to sorption of functional sulfogroups of the KU 2-8 resin decreased along with the sodium ion content increase. In the solutions of a sodium concentration of 0.5 mol/L, the sorption of Sr-90 proceeded mainly on the phase of iron oxide immobilized in the polymer matrix. In Figure 6, the *K_d_* values of Sr-90 were converted to the calculated mass of iron oxide (Table 2), i.e., without taking into account the mass of the polymer, as compared to the pure powders.

Since the hematite-based composites were less efficient at removing Sr-90 radionuclides, further work was performed with the magnetic sorbent KU–Mag. The results of evaluating the effect of the V/m ratio on the sorption value, as well as Sr-90 K_d_ in various model solutions, including those simulating the clarified part of heterogeneous LRW, are shown in Figure 7.

Figure 8 shows the output sorption and desorption curves obtained using the KU–Mag sorbent. According to the sorption curves (Figure 5a), a decrease in DF down to 10 occurred when more than 100 bed volumes of the model solution No. 2 were fed. After the third cycle, the volume of the model solution, filtered before the breakthrough, decreased. However, in comparison with the first cycle, the decrease in resource was insignificant, which indicated the stability of the sorbent. Using the HNO_3_ solution as an eluent allowed more than 99% of Sr-90 in the both sorption cycles to be desorbed and made it possible to reuse the material.

## 4. Discussion

The requirement to immobilize iron oxide in the polymer matrix is related to the fact that pure powders exhibit low mechanical strength and cannot be used under the dynamic sorption conditions. In addition, removing such powders from the solutions to be decontaminated is also related to certain difficulties. Immobilization of the iron oxide phase in the polymer matrix allowed mechanically strong sorbents to be obtained. The formation of the iron oxide phase in the polymer matrix was followed by reduction of the particle size by more than two times compared to pure powders (Table 3). This fact can be explained by the effect of the polymer network limiting the growth of particles during the hydrothermal treatment.

In the neutral solutions of NaNO_3_ (Figure 4a), the sorption-selective characteristics of the original KU 2-8 and the composite sorbents were identical, while the pure iron oxide powders sorbed Sr-90 to a lesser degree, which was especially noticeable in the solutions with low mineralization (10^−3^–10^−1^ mol/L). This was because of the fact that in the solutions with low concentrations of sodium ions the adsorption of Sr-90 proceeded mainly due to the ion exchange on the functional sulfogroups of the KU 2-8 resin. As a result, the adsorption was effective both on the initial KU 2-8 and on the composites. Here, in neutral media, pure iron oxide powders did not virtually sorb Sr-90.

At switching to the alkaline solutions (pH 13), on the contrary, the selectivity of the composite sorbents and iron oxide powders increased dramatically (Figure 4b), which complied with the results provided in [8]. Here, hematite and magnetite powders demonstrated the best sorption-selective characteristics. The sorption of radionuclides on the KU 2-8 resin decreased sharply when the concentration of NaNO_3_ grew from 0.1 to 0.5 mol/L, which was determined by the low selectivity of the cation-exchanger despite its high sorption capacity. In the solutions with a concentration of NaNO_3_ of 0.5 mol/L and above, KU 2-8 was not effective, so that the possibility of adsorption of the radionuclide due to ion exchange on SO_3_H groups in highly mineralized solutions can be neglected.

A noticeable increase in the removal efficiency towards Sr-90 on the composite sorbents and iron oxide powders was observed at pH 11 and above (Figure 5). This increase in the efficiency was presumably related to the fact that the charge on the composite sorbent surface was converted to negative, which promoted the sorption of Sr^2+^ ions [21]. The advanced adsorption on the magnetic samples MAG and KU–Mag was in agreement with the data provided by Todorović et al. [29], according to which the strontium sorption proceeded to a greater degree on magnetite rather than on hematite. The authors suggested that this was related to the increased strength of the positive repulsive field of Fe(III) for the hematite compared to Fe(II) of the magnetite [29].

The *K_d_* values for Sr-90 recalculated for the mass fraction of iron oxide in the composite sorbent (Figure 6) exceeded those obtained for the pure magnetite and hematite powders. Based on this fact, one can conclude that the formation of the iron oxide phase in the polymer matrix did not affect the extraction efficiency towards Sr-90, but, on the contrary, led to the improvement in the sorption-selective characteristics, probably, due to the decrease of peptization.

Based on the results obtained, further work was performed with the composite sorbent KU–Mag, whose sorption properties were evaluated in the model solutions (model solution Nos. 3, 4), simulating the clarified part of heterogeneous LRW at varied V/m ratios (Figure 7). With the increasing V/m, there was a naturally following decrease in the sorption efficiency towards Sr-90: here, the highest efficiency was observed for the model solution No. 3. At low V/m (50–100 mL/g), the sorption values exceeded 80%, which indicated that this material was promising for Sr-90 extraction. The values of Sr-90 K_d_ remained almost within the same order at changing V/m, which indicated high selectivity of the KU–Mag sorbent towards the radionuclide in the presence of Na^+^ ions. The sorption-selective characteristics of the KU–Mag sorbent in the model solution No. 2 were reduced, due to the lowest NaOH content compared to the solutions No. 3 and 4. Based on this fact, one can conclude that the sorption efficiency was largely determined by the pH of the solution rather than by mineralization.

The sorption properties of KU–Mag were evaluated under dynamic sorption conditions. During the experiment, we did not observe any mechanical destruction of the sorbent, which indicated that the hydrothermal treatment at 175 °C did not affect the strength of the polymer crosslinking. Neither changes in the filtrate color nor formation of a sediment due to elution of iron were recorded.

In the work of Granizo et al. [30] devoted to the study on sorption of Cs-137 on magnetite particles, it was found that a part of cesium was irreversibly sorbed, which fact the authors associated with the formation of Fe–Cs spinels. However, in our case, this phenomenon was not observed for Sr-90. When feeding 50 column volumes of HNO_3_ solution of a concentration of 1 mol/L, the desorption efficiency in both sorption cycles exceeded 99%.

## 5. Conclusions

The composite sorbents for extraction of Sr-90 radionuclide from highly mineralized alkaline media were synthesized by immobilization of the magnetite or hematite phase in the polymer matrix of the ion exchange resin KU 2-8. The magnetite-based sorbents appeared to be more promising as compared to the hematite-based sorbents due to their greater sorption efficiency and the possibility of removal from the solution to be decontaminated by magnetic separation. In the model solution that simulated the clarified part of heterogeneous LRW, the Sr-90 K_d_ value for the magnetic composite equaled to 10^3^–10^4^ mL/g, depending on the V/m ratio, which indicated high selectivity and promising features of this material. Under the dynamic conditions, the sorbent appeared to be able to decontaminate more than 100 bed volumes from Sr-90 with DFs of more than 10. It has been demonstrated that the reuse of the material is possible after desorption of the accumulated radionuclide with a solution of the 1 mol/L HNO_3_.

## Figures and Tables

**Figure 1 materials-13-01189-f001:**
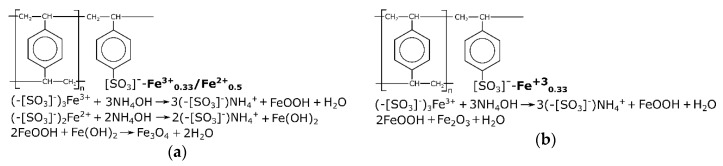
Structure of the KU 2-8 ion-exchanger and the reaction for producing oxides in the phase of the resin: (**a**) KU–Mag; (**b**) KU–Hem.

**Figure 2 materials-13-01189-f002:**
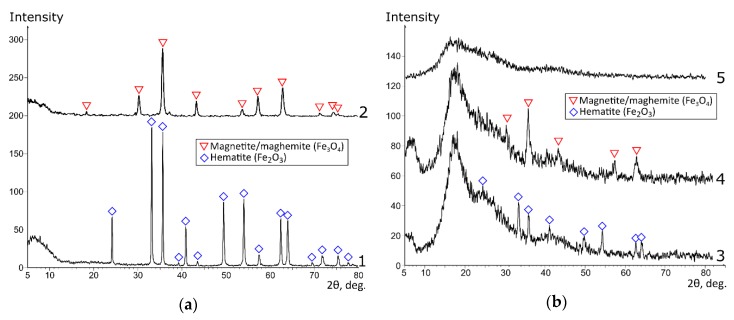
The X-ray patterns of the samples: (**a**) original iron oxides; (**b**) composite sorbents; 1—Hem, 2—Mag, 3—KU–Hem, 4—KU–Mag, 5—KU 2-8 in the Na form.

**Figure 3 materials-13-01189-f003:**
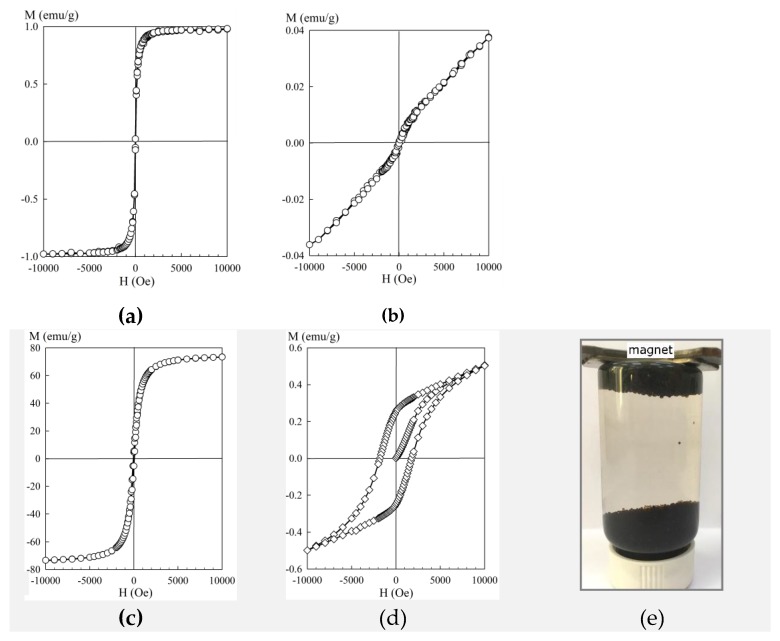
Hysteresis loops of the sorption materials: (**a**) KU–Mag; (**b**) KU–Hem; (**c**) Mag; (**d**) Hem; and (**e**) magnetic properties of KU–Mag.

**Figure 4 materials-13-01189-f004:**
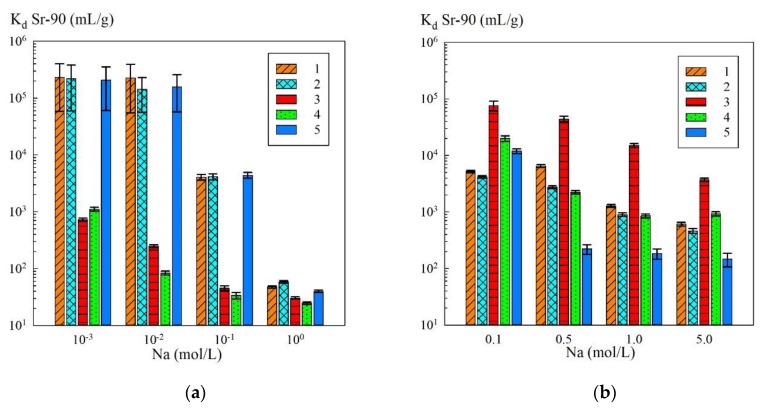
Dependence of K_d_ Sr-90 on the concentration of Na^+^ ions: (**a**) neutral medium (NaNO_3_); (**b**) alkaline medium pH 13 (NaNO_3_ + NaOH); 1—KU–Mag, 2—KU–Hem, 3—Mag, 4—Hem, 5—KU 2-8 in Na-form.

**Figure 5 materials-13-01189-f005:**
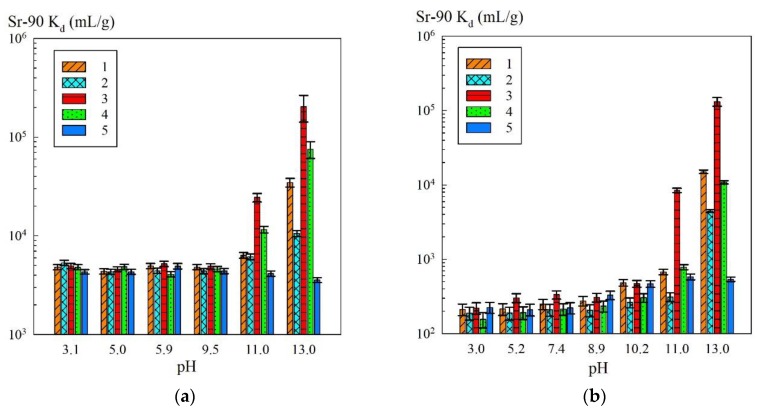
Dependence of K_d_ Sr-90 on the pH of the solution: (**a**) Na^+^—0.1 mol/L (NaNO_3_ + NaOH); (**b**) Na^+^—0.5 mol/L (NaNO_3_ + NaOH); 1—KU–Mag, 2—KU–Hem, 3—Mag, 4—Hem, 5—KU 2-8 in Na-form.

**Figure 6 materials-13-01189-f006:**
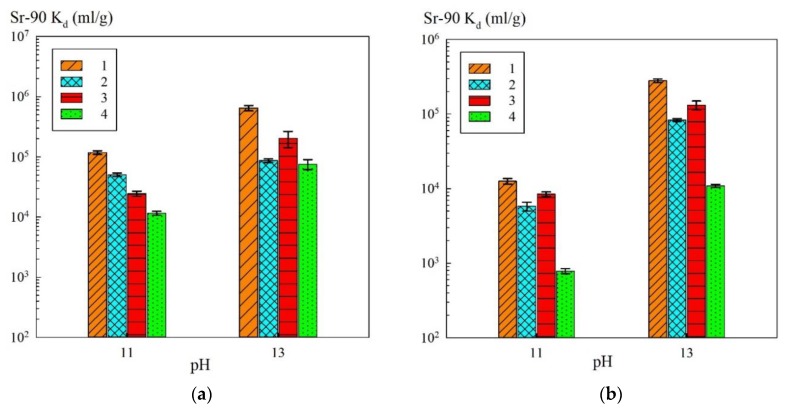
Diagrams of the dependence K_d_ to Sr-90 on pH in the equivalent of iron oxide: (**a**) Na^+^—0.1 mol/L (NaNO_3_ + NaOH); (**b**) Na^+^—0.5 mol/L (NaNO_3_ + NaOH); 1—KU–Mag, 2—KU–Hem, 3—Mag, 4—Hem.

**Figure 7 materials-13-01189-f007:**
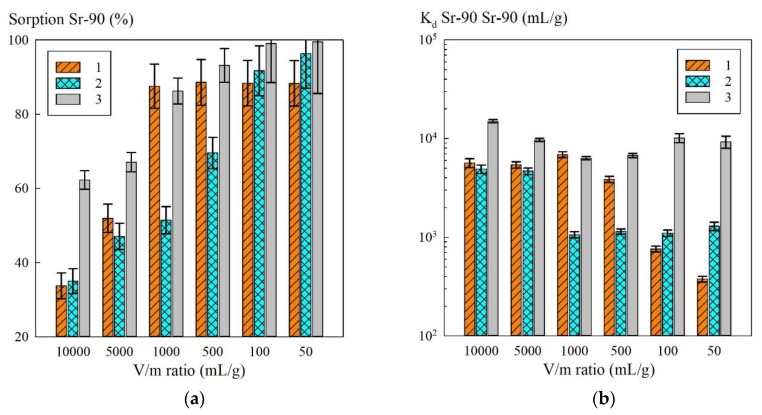
Sorption-selective characteristics of the KU–Mag sorbent at different V/m ratios: (**a**) dependence of sorption on the V/m ratio; (**b**) dependence of K_d_ to Sr-90 on the V/m ratio; 1—model solution No. 2, 2—model solution No. 3, 3—model solution No. 4.

**Figure 8 materials-13-01189-f008:**
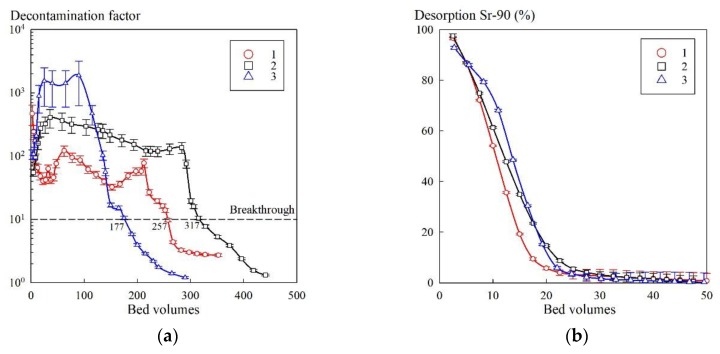
Adsorption–desorption characteristics of the KU–Mag sorbent under dynamic conditions: (**a**) adsorption of Sr-90 from the solution model No. 2; (**b**) desorption of Sr-90 using HNO_3_ solution of 1 mol/L.

**Table 1 materials-13-01189-t001:** Composition of model solutions used in the work.

Components of Solution	Concentration (mol/L)			
	Solution No. 1	Solution No. 2	Solution No. 3	Solution No. 4
NaNO_3_	-	0.4	2.25	0.484
NaOH	0.1	0.1	0.75	3.0
Na_2_SO_4_	-	-	-	0.16
NaCl	-	-	-	0.28
Na_2_SiO_3_	-	-	-	0.0089
K_2_CrO_4_	-	-	-	0.023
Al(NO_3_)_3_	-	-	-	0.22
NaNO_2_	-	-	-	0.761

**Table 2 materials-13-01189-t002:** Content of iron and iron oxides in the composite sorbents.

Sorbent	Content of Fe (mg/g)	Crystalline Phase	Content of the Crystalline Phase (wt %)
KU–Hem	43	Hematite (Fe_2_O_3_)	12.1
KU–Mag	122	Magnetite (Fe_3_O_4_)	5.3

**Table 3 materials-13-01189-t003:** Particle sizes of iron oxides for the pure powders and composites.

Hem	Angle 2θ	24	33	36	41	50	54	63	64
	Particle size (nm)	35	35	38	34	28	26	27	25
	Average size (nm)	31							
KU–Hem	Angle 2θ	-	33	36	-	50	54	63	64
	Particle size (nm)	-	8	14	-	14	16	10	17
	Average size (nm)	13							
Mag	Angle 2θ	30	35	43	54	57	63	-	-
	Particle size (nm)	18	18	30	62	19	18	-	-
	Average size (nm)	28							
KU–Mag	Angle 2θ	-	36	44	-	58	63	-	-
	Particle size (nm)	-	10	6	-	9	13	-	-
	Average size (nm)	10							

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
