# Peer review of "Synthesis and Sorption Properties towards Sr-90 of Composite Sorbents Based on Magnetite and Hematite"

_materials, 2020, doi:10.3390/ma13051189_

Round 1
Reviewer 1 Report
Comments
The issues covering the article material are very important. Radionuclides, or radioactive isotopes - varieties of elements whose nuclei are unstable and undergo radioactive transformation, including Strontium-90. Strontium-90 (90Sr) is a radioactive isotope of strontium produced by nuclear fission, with a half-life of 28.8 years. It undergoes β− decay into yttrium-90, with a decay energy of 0.546 MeV. Strontium-90 has applications in medicine and industry and is an isotope of concern in fallout from nuclear weapons and nuclear accidents. Therefore, any research activity regarding the creation of efficient materials for the treatment of liquid radioactive waste is highly recommended.
Research topic “Synthesis and Sorption Properties towards Sr-90 of Composite Sorbents Based on Magnetite and Hematite” is important due to the use of a composite - iron oxide in a polymer matrix, instead of powders that show low mechanical strength under dynamic sorption and are difficult to remove from sanitized solutions.
The aim of the study, which was the synthesis and study of sorption characteristics of sorbents based on iron oxides - magnetite and hematite, immobilized in the mass of an ion exchange resin polymer, with subsequent hydrothermal treatment, was achieved. The polymer-iron oxide composite allowed to increase the durability of oxides, reduce their volume and avoid aggregation. It has been shown that sorbent based on magnetite is the most efficient (over 95%) - it showed the highest selectivity in relation to Sr-90, it is suitable for use in conditions of dynamic sorption with subsequent desorption of radionuclide with HNO3 solution.
The research methodology and reliability of the results is beyond doubt - research conducted in a wide range of specialists from various research centers, using specialized equipment.
An additional advantage of this methodology is the use of relatively cheap materials.
I suggest introducing some slight corrections regarding the presentation of results in Figures 4, 5, 6 and 7. Namely, instead of the results presented on a logarithmic scale in the form of bars, present the results in the form of curves based on measurement points, or in the form of properly selected non-linear functions. It seems to me that such an illustration would be more transparent to the reader.

Author Response
We are grateful to reviewer for the positive feedback to our article.
We totally agree with reviewer, that the results presentation using symbols instead of diagrams, it is the easiest method. We tried to show data in curves, but we were not able to get a satisfactory result. The several experimental values could be different more than on one order. It complicates their interpolation or using a nonlinear function. In some cases, the experimental data could overlap; it complicates to display the results. Based on this we left the previous figures and add confidence intervals.
Reviewer 2 Report
The Authors present synthesis and sorption properties of some polymer composite based on hematite and magnetite. An importance and novelty of this research should be better highlighted by the Authors in the view of recent research described at least in Journal of Hazardous Materials. Nothing is said about experimental error and statistical analysis of the experimental data (introducing of some statistical scattering on the resulting graphs). A short discussion of parameter sensitivity of the process would be also important. It would be interesting for the readers to read about possible engineering applications also.
Author Response
We are grateful to reviewer for valuable comments, to make our article better, and gave us a positive feedback. The answers on remarks are given below:
Answers
1. The figures was changed, and the confidence interval values were added.
2. Line 92. Following additions was entered in article text. The confidence interval for experimental values with confidence probability 0.95 (standard deviation × 1.96) is calculated and given below:
For a series of parallel experiments (sampling) we used one-sample t-test.
For cases when using of sampling is impossible (Figure 8) the confidence interval was calculated with formula (6).
3. Line 185. ). Minimum detectable activity is ~ 1 Bq/L with natural background 1 decay per minute. The measurement error depending of extinction coefficient (blanking ratio), is 5%.
4.The links No. 18, 25 were added, were published in Journal of Hazardous Materials.
Line 53. The perspective sorbent was synthesized for extraction arsenic from liquid media. The material is particles of iron oxide, which homogeneous dispersed, in the acrylamide-based cryogel [18]. Thereby, the material has stability and high adsorption capacity
Line 71. From a technical point of view, the most suitable is the composite material presented in the work [25]. The sorbent, which is Fe3O4 nanoparticles immobilized by penicillium sp mycelium, can be used to extract Sr(II), Th(IV) and U(VI) from liquid media. The composite material has high stability and adsorption capacity and can be used in sorption-desorption cycles. Despite the merits, the composites synthesis is quite difficult and includes mycelium growing penicillium sp during a long time. Besides, authors did not give a data about using this material in dynamic sorption conditions.
5. Received composites can be used for purification of the clarified portion of alkaline heterogeneous LRW, it is indicated in Conclusion.

Reviewer 3 Report
This manuscript describes the synthesis of a series of composite sorbents including cation-exchanger and either magnetite or hematite for the recovery of 90Sr from highly saline industrial solutions. This interesting work deserves publication in Materials. However, some sections could be improved and some questions should be addressed to reinforce the impact of the work.
- Editing
The manuscript would require some improvement. It is not common citing: “The authors of [9] …” (in several places). There are also some typing mistaked to be corrected along the manuscript.
The type, nature or characteristics of the polymer used for encapsulating mineral phases are not reported in the abstract. This could be helpful briefly stating this information.
- Materials and methods
The section could be re-organized in sub-sections (materials, synthesis of magnetite, synthesis of hematite, synthesis of magnetite/composite, synthesis of hematite magnetite, characterization, experimental sorption procedures, analysis). This would make clearer the description of synthesis procedures, which are, under the current form, debatable and questionable.
- Table 2:
Could the authors clearly specify how they calculate the effective content of the crystalline phase (wt. %) (right column).
- Magnetization
The magnetization of the composite magnetite material is quite low compared to literature. How the authors explain this weak value? The weak amount of magnetite core? They claim that this is beneficial due to preventing agglomeration. But is it sufficient for efficient magnetic separation?
- Encapsulation effects
The encapsulation of the magnetite core with polymer means possible limitations for mass transfer through resistance to intra-particle diffusion. Investigating batch kinetics would help in identifying this diffusion effects. As far as I understand, the effective sorption of strontium under the LRW conditions (highly saline alkaline solutions) is essentially occurring on the magnetite core of the composite (and much less on the polymer resin compartment). This means that the very limited fraction of inorganic compartment considerably limits the potential of the composite sorbent in terms of both equilibrium (thermodynamics) and kinetics (diffusion effects). The selection of experimental conditions should then be justified (benefit of polymer for the specific target application of LRW treatment).
- Stability of material under irradiation
This interesting work appeals supplementary question on the evaluation of the stability of the composite when submitted to irradiation. Did the authors check the behavior of their composite material under irradiation? Did they check the changes in the chemical structure/physical structure of their materials after being in contact with several active batches (recycling) by FTIR etc… This is not under the focus of this work but discussing this question would be attractive and appealing for readers.
Author Response
We are grateful to reviewer for the valuable and important comments, that allowed us to improve the article. And we are also grateful for the positive feedback.
- Editing. The manuscript would require some improvement. It is not common citing: “The authors of [9] …” (in several places). There are also some typing mistaked to be corrected along the manuscript.
Answer
Line 42. «…The authors of…» was changed to «…Karasyova et al…».
Line 57. «…The authors of…» was changed to «…Hansen et al…».
- The type, nature or characteristics of the polymer used for encapsulating mineral phases are not reported in the abstract. This could be helpful briefly stating this information.
Answer
Monomer of polymer structure present in Figure 1.
Line 77. The short information about ion-exchange resins structure was added.
- Materials and methods. The section could be re-organized in sub-sections (materials, synthesis of magnetite, synthesis of hematite, synthesis of magnetite/composite, synthesis of hematite magnetite, characterization, experimental sorption procedures, analysis). This would make clearer the description of synthesis procedures, which are, under the current form, debatable and questionable.
Answer
In «Materials and Methods» were added the according subsections.
- Table 2. Could the authors clearly specify how they calculate the effective content of the crystalline phase (wt. %) (right column).
Answer
Line 144. Some explanation was added.
- Magnetization. The magnetization of the composite magnetite material is quite low compared to literature. How the authors explain this weak value? The weak amount of magnetite core? They claim that this is beneficial due to preventing agglomeration. But is it sufficient for efficient magnetic separation?
Answer
Really, the low magnetization is connecting with low magnetite in sorbent composition. However, the magnetic composite can be separated from the solution using magnetic separation. As an evidence, we added the figure 3e.
- Encapsulation effects. The encapsulation of the magnetite core with polymer means possible limitations for mass transfer through resistance to intra-particle diffusion. Investigating batch kinetics would help in identifying this diffusion effects. As far as I understand, the effective sorption of strontium under the LRW conditions (highly saline alkaline solutions) is essentially occurring on the magnetite core of the composite (and much less on the polymer resin compartment). This means that the very limited fraction of inorganic compartment considerably limits the potential of the composite sorbent in terms of both equilibrium (thermodynamics) and kinetics (diffusion effects). The selection of experimental conditions should then be justified (benefit of polymer for the specific target application of LRW treatment).
Answer
Really, in the solutions with high mineralization, adsorption pass on the magnetite, functional groups of ion-exchange resin is not effective due to low selectivity. Obtaining of any sorption composite sorbent leads to decreasing mass of sorption active component. It leads to some efficiency decreasing in calculating to the total mass of the composite. It is confirmed in Figures 4 and 5. This is the common disadvantage of the most number composite sorbents. However, the synthesis of composite sorbents can achieve new characteristics, for example, the possibility of using in dynamic condition of sorption.
About the influences polymer on diffusion and sorption equilibrium, we can assume this: the kinetic parameters will be determined intraparticle diffusion rate. The resin KU 2-8 is gel type ion exchanger, that’s why the polymer matrix is permeable to cations. The polymer matrix does not influence on sorption equilibrium, this is indirectly confirmed by results on Figure 6.
Line 77-82. The some explanations and links 26 and 27 were added.
- Stability of material under irradiation. This interesting work appeals supplementary question on the evaluation of the stability of the composite when submitted to irradiation. Did the authors check the behavior of their composite material under irradiation? Did they check the changes in the chemical structure/physical structure of their materials after being in contact with several active batches (recycling) by FTIR etc… This is not under the focus of this work but discussing this question would be attractive and appealing for readers.
Answer
We completely agree with reviewer about necessity of future the composite radiolytic stability research. Unfortunately, at this moment we do not have a powerful ionizing radiation source, which can form an absorbed dose at least tens of Gy.

Round 2
Reviewer 2 Report
The Authors have inserted some relatively fresh references concerning a matter presented in this article, nevertheless some linguistic revision is still recommended.
Author Response
Some changes are added to the text.
Reviewer 3 Report
The authors answered most of the questions. I fully understand the answer to my last question (missing opportunity to get access to irradiation source), however, the FTIR analysis after 3 recyclings could have been performed to verify the stability of the sorbent.
I suggest acceptiong this revised version of the manuscript.
Author Response
We apologize for not being able to fully respond to the comments of the Reviewer. According to idealized calculations (taking into account the geometry and integral accumulation of the radionuclide throughout the volume), the KU 2-8 MAG material has accumulated a dose of 0.01-10 Gy during the experiment. There is information in the literature that noticeable changes in ion exchangers, such as decreased ion-exchange capasity, are observed at doses greater than 105-106 Gy (Pillay, K.K.S. A review of the radiation stability of ion exchange materials. Journal of Radioanalytical and Nuclear Chemistry, Articles. 1986, 102 247-268. Doi: 10.1007 / BF02037966). In this case, it can be assumed that FTIR will not allow detecting noticeable changes in the structure of the polymer due to the low absorbed dose.